# Blood transfusion and the risk for infections in kidney transplant patients

**David Massicotte-Azarniouch**[1], **Manish M. Sood**[2], **Dean A. Fergusson**[3], **Michaël Chassé**[4], **Alan Tinmouth**[5], **Greg A. Knoll**[2]*

1 Division of Nephrology, Department of Medicine, University of Ottawa, Ontario, Canada, 2 Division of Nephrology, Department of Medicine, Clinical Epidemiology Program, Ottawa Hospital Research Institute, University of Ottawa, Ontario, Canada, 3 Department of Medicine, Clinical Epidemiology Program, Ottawa Hospital Research Institute, University of Ottawa, Ontario, Canada, 4 Department of Medicine, University of Montreal, Montreal, Quebec, Canada, 5 Division of Hematology, Department of Medicine, University of Ottawa, Ottawa, Ontario, Canada

* gknoll@toh.ca

## Abstract

### Background

Receipt of a red blood cell transfusion (RBCT) post-kidney transplantation may alter immunity which could predispose to subsequent infection.

### Methods

We carried out a single-center, retrospective cohort study of 1,258 adult kidney transplant recipients from 2002 to 2018 (mean age 52, 64% male). The receipt of RBCT post-transplant (468 participants transfused, total 2,373 RBCT) was analyzed as a time-varying, cumulative exposure. Adjusted cox proportional hazards models were used to calculate hazard ratios (HR) for outcomes of bacterial or viral (BK or CMV) infection.

### Results

Over a median follow-up of 3.8 years, bacterial infection occurred in 34% of participants at a median of 409 days post-transplant and viral infection occurred in 25% at a median of 154 days post-transplant. Transfusion was associated with a step-wise higher risk of bacterial infection (HR 1.35, 95%CI 0.95–1.91; HR 1.29, 95%CI 0.92–1.82; HR 2.63, 95%CI 1.94–3.56; HR 3.38, 95%CI 2.30–4.95, for 1, 2, 3–5 and >5 RBCT respectively), but not viral infection. These findings were consistent in multiple additional analyses, including accounting for reverse causality.

### Conclusion

Blood transfusion after kidney transplant is associated with a higher risk for bacterial infection, emphasizing the need to use transfusions judiciously in this population already at risk for infections.

**Data Availability Statement:** Data cannot be shared publicly because of ethical concerns and privacy restrictions in accordance with our

institution's ethics board. Data requests can be directed to the Ottawa Health Sciences Network Research Ethics Board (rebadministration@ohri. ca) or holiu@ohri.ca.

**Funding:** This project was funded by a Canadian Blood Service Infrastructure Grant.The funders had no role in study design, data collection and analysis, decision to publish, or preparation of the manuscript.

**Competing interests:** The authors have declared that no competing interests exist.

**Abbreviations:** CAKUT, congenital anomalies of the kidneys and urinary tract; CVD, cardiovascular disease; DGF, delayed graft function; ESKD, end-stage kidney disease; GN, glomerulonephritis; HR, hazard ratio; ICD10, International Statistical Classification of Diseases and Related Health Problems; IQR, inter-quartile range; OHDW, Ottawa Hospital Data Warehouse; PCKD, polycystic kidney disease; PRA, panel reactive antibodies; RBCT, red blood cell transfusion; SD, standard deviation; TOH, The Ottawa Hospital.

## Introduction

Kidney transplantation is the treatment of choice for end-stage kidney disease (ESKD) since it is associated with improved survival and quality of life compared to dialysis [1–3]. Transplantation comes with certain risks, one of which is the development of anemia after transplant. Indeed, 40–70% of kidney transplant patients require a red blood cell transfusion (RBCT) during their post-transplant course for the treatment of anemia [4–7], due to surgical blood losses, delayed graft function (DGF) and bone marrow suppression from immunosuppressive medications [8–10]. A RBCT is, however, not a benign intervention as it may have important immunomodulatory effects relevant to kidney transplant patients.

In the non-transplant population, peri-operative RBCTs are associated with a higher infection risk [11–13]. It is thought that a RBCT could contain immune suppressing substances which could impair cellular immunity and induce a state of anergy in the recipient [14–17]. This immune down regulation was a key rationale for the use of pre-transplant blood transfusion to improve allograft outcomes, a practice that has fallen out of favour with more potent immunosuppressants and the recognized risk for sensitization [18]. However, this potential immunosuppressive effect merits further clarification in kidney transplant patients given their high exposure to RBCT, their immunosuppressed status and high incidence of infection [19].

The objective of this study was to examine the association of infection (bacterial or viral) with post-kidney transplantation receipt of RBCT. We hypothesized that RBCT would be associated with a higher risk of infection.

## Materials and methods

### Study design, participants and setting

This study was part of a larger project by our group examining the risks from blood transfusions in kidney transplant recipients. Therefore, its methods are similar, except for the outcomes, as those reported by our group in a recently published study on the risks for adverse graft outcomes associated with RBCT [20]. We present here the association of RBCT and infectious outcomes. All adult kidney transplant recipients at The Ottawa Hospital (TOH), from January 1st, 2002 until December 31st, 2018 were included in this retrospective cohort study. As a tertiary academic facility, TOH provides services for Eastern Ontario and Western Quebec (catchment area 1.2 million) with a 1,200 bed, 3-campus teaching hospital. Only kidney transplant recipients were included as kidney-other organ transplants are not performed at our center. S1 File details the usual immunosuppression protocol and used at our center. The study start date is January 1st 2002 which aligns with the adoption of the 10th revision of the International Statistical Classification of Diseases and Related Health Problems (ICD10) at our institution and is the start of transfusion data being captured. All elements of the study design, including exposure, outcomes and analytic plan were devised prior to data retrieval. The Ottawa Hospital Research Institute, our institutional research and ethics board, approved the study with waived consent due to the retrospective nature of our study. The reporting of this manuscript follows the RECORD (extension of STROBE) guidelines for observational studies using routinely-collected health data (S1 Table) [21].

### Data sources

The TOH Renal Transplant database, a prospectively collected, monthly updated database, was used to identify study individuals. The main exposure, outcomes and baseline characteristics were identified through the Ottawa Hospital Data Warehouse (OHDW), a data repository of routinely collected health administrative data on all patients treated at all campuses of TOH.

Additional information on baseline characteristics and outcomes were extracted by chart review. A unique encoded patient identifier was used for data linkage.

## Exposure

Receipt of a RBCT after kidney transplant (post-operative day 1 onwards) was treated as the main study exposure. Since we did not have the exact time of the transplant surgery, we could not know if a RBCT given on the day of the surgery was before or after transplant. At our institution, there is no strict hemoglobin level required for transplant to proceed and the need for pre-transplant transfusion is determined amongst the transplant surgery, nephrology and anesthesia services. The time for the start of exposure was defined as the date and time of issuing of RBCT from our institution's blood bank. All RBCTs at TOH were leukodepleted during the study period as per our routine practice. Nephrologists provide the care for kidney transplant recipients in consultation with the surgical team. When a non-urgent RBCTs is required, confirmation is typically required with the Nephrology service prior to receipt. Ultimately, receipt of a transfusion is left to the discretion of the clinician with decision-making based on regional clinical transfusion guidelines [22].

## Outcomes

The outcomes of interest were: 1) bacterial infection; and 2) viral infection. Bacterial infection was defined as either: a) bacteremia (any positive blood culture); b) pneumonia (ICD10 discharge code for pneumonia during a hospital encounter); c) sepsis (ICD10 discharge code for sepsis during a hospital encounter); d) urinary tract infection (positive urine culture occurring during a corresponding hospital encounter with an ICD10 discharge code for acute pyelonephritis or urinary tract infection); or e) positive clostridium difficile stool toxin test. See S2 Table for details on ICD codes. The ICD codes for pneumonia, sepsis and pyelonephritis have previously been validated (positive predictive values 83 to 97%) [23]. Viral infection was defined as either: a) BK virus infection (BK DNA PCR viral load >500U/mL in at least 2 measurements within at least 1 month of each other, or histologically confirmed BK nephropathy on kidney biopsy); or b) CMV infection (CMV DNA PCR viral load >137 IU/mL on one occasion, or histologically confirmed tissue-invasive CMV infection). BK and CMV viral load data only became available in our databases from February 1st, 2007 and March 14th, 2014, respectively. Therefore, for analysis of viral infection, the cohort was limited to those receiving a kidney transplant from March 14th, 2014 onwards. The definitions for BK and CMV viremia were based on cut-offs used at our institution. We routinely screen for BK viremia during the first year after transplant (monthly until 6 months then at 9 months and at 12 months). Prophylaxis against CMV with valganciclovir is given for 6 months to those at high risk (donor positive recipient negative) and for 3 months to CMV positive patients who receive lymphocyte depleting induction (thymoglobulin at our center).

## Baseline demographics and covariates

We gathered baseline patient characteristics available in our data sources, determined a priori to be clinically relevant. S3 Table lists these variables and the data sources used. Baseline characteristics were from the day of kidney transplant.

## Statistical analysis

Descriptive statistics (means with standard deviations [SD], medians with inter-quartile ranges [IQR] and counts with percentages) were presented along with appropriates test statistic for each covariate based on being transfused or not post-transplant.

A time-to-event analysis was used to analyze the association between RBCT and outcome. The index date (time 0) for the start of follow-up for each observation was the date of kidney transplant. Each observation was followed until occurrence of an outcome or a censoring event (graft loss [date of return to permanent dialysis], loss to follow-up [transfer to another program, captured in the Renal Transplant Database], death, or end of study period [31st December 2018]). The main exposure, RBCT, was analyzed as a time-dependent, cumulative exposure [24, 25]; each observation became exposed upon receipt of a RBCT and for every additional RBCT their cumulative exposure increased accordingly [26, 27].

Crude cumulative incidences were calculated for the study outcomes. Cox proportional hazard models were used to estimate the cause-specific hazard ratio (HR) for the study outcomes, adjusted for clinically relevant baseline covariates. Diabetes and DGF were not included because they were co-linear with the variables "Cause of ESKD" (which includes diabetes) and "t-cell depleting induction" (DGF is considered high-immunological risk at our institution and such patients receive thymoglobulin), respectively. As the functional form of cumulative RBCT was non-linear, exposure was categorized as "None", "1 RBC", "2 RBC", "3–5 RBC" and ">5 RBC" to provide the most equal distribution of groups among transfused individuals. All statistical analyses were done using SAS version 9.4 (SAS Institute Inc., Cary NC, USA).

## Additional analyses

We also performed the following analyses: 1) accounting for reverse causality (where the outcome leads to the exposure of interest) by examining time lags of 3, 7, 10 and 14 days between RBCT and an outcome for the RBCT to be considered an exposure since acute infection may lead to anemia and it could take days to weeks for an immunosuppressive effect to manifest itself; 2) association of RBCT with any infection (both bacterial and viral; cohort limited to March 14, 2014 onwards); 3) stratified by DGF status since, as explained above, we did not include DGF in our adjusted analyses but an individual with DGF could be at greater risk of both requiring a RBCT and infection; 4) to account for potential misclassification due to RBCT on the day of transplant not being counted, we re-conducted our analyses where 50% of those transfused on the day of transplant were assigned as receiving a RBCT immediately post-transplant; 5) adjusting for the year of kidney transplant to account for changes in transfusion and transplantation practices; 6) association of RBCT with infection, whether it occurred after an acute rejection episode or not, since rejection may be accompanied by anemia and will often be treated with increased immunosuppression which may predispose to infection; 7) after noticing post-hoc a long time-frame between occurrence of first transfusion and of infection, we performed our analyses restricting to the first 90-days post-transplant.

## Results

### Patient and transfusion characteristics

The patient and transfusion characteristics are consistent with what was previously reported by our group [20]. Our study population consisted of 1,258 kidney transplant recipients with a median follow-up of 3.8 years. There were 468 (37.2%) who were transfused and these patients were older, more often female, had more co-morbidities (diabetes, cardiovascular disease) and more often received T-cell depleting induction therapy compared to those not transfused. (Table 1). A total of 2,373 RBCT were given throughout the study period (incidence of 33 RBCT per 100 person-years) the majority of which occurred in the first week post-transplant. (Table 2). There was a decrease in the one-year incidence of RBCT over time (Fig 1). The mean hemoglobin level pre-RBCT was 7–8 g/dl and this progressively decreased throughout the study period (S1 Fig).

**Table 1. Baseline characteristics.**

| Characteristic | Total cohort | Never transfused | Transfused during study | p-value |
|---|---|---|---|---|
| # of transplant recipients (%) | 1,258 (100) | 790 (62.8) | 468 (37.2) | |
| Age; mean (SD) | 52 (14) | 50.6 (13.9) | 54.0 (14.4) | <0.0001 |
| Female; no. (%) | 454 (36.1) | 235 (29.8) | 219 (46.8) | <0.0001 |
| Living donor transplant; no. (%) | 571 (45.4) | 417 (52.8) | 154 (32.9) | <0.0001 |
| Race; no. (%) | | | | 0.37 |
| • Caucasian | 967 (76.8) | 614 (77.7) | 353 (75.4) | |
| • Black | 109 (8.7) | 67 (8.5) | 42 (9.0) | |
| • Asian | 68 (5.4) | 35 (4.4) | 33 (7.1) | |
| • Middle-Eastern | 57 (4.5) | 37 (4.7) | 20 (4.3) | |
| • Other | 57 (4.5) | 37 (4.7) | 20 (4.3) | |
| Cause of ESKD; no. (%) | | | | 0.0026 |
| • GN | 413 (32.8) | 283 (35.8) | 130 (27.8) | |
| • Diabetes | 315 (25.0) | 182 (23.0) | 133 (28.4) | |
| • PCKD | 180 (14.3) | 122 (15.4) | 58 (12.4) | |
| • CAKUT | 101 (8.0) | 64 (8.1) | 37 (7.9) | |
| • Other | 249 (19.8) | 139 (17.6) | 110 (23.5) | |
| Comorbidity; no. (%) | | | | |
| • Diabetes | 389 (30.9) | 221 (28.0) | 168 (35.9) | 0.0033 |
| • CVD* | 251 (20.0) | 134 (17.0) | 117 (25.0) | 0.0006 |
| Kidney transplant number; no. (%) | | | | 0.52 |
| • 1 | 1161 (92.3) | 733 (92.8) | 428 (91.5) | |
| • 2 | 87 (6.9) | 52 (6.6) | 35 (7.5) | |
| • 3 | 8 (0.6) | 4 (0.5) | 4 (0.9) | |
| • 4 | 1 (0.1) | 1 (0.1) | 0 (0) | |
| • 5 | 0 (0) | 0 (0) | 0 (0) | |
| • 6 | 1 (0.1) | 0 (0) | 1 (0.2) | |
| Previous non-kidney transplant; no. (%) | 14 (1.1) | 10 (1.3) | 4 (0.9) | 0.50 |
| Recipients transplanted more than once during study period; no. (%) | 34 (2.7) | 19 (2.4) | 15 (3.2) | 0.40 |
| PRA; no. (%) | | | | 0.63 |
| • 0% | 590 (47) | 381 (48.2) | 209 (44.7) | |
| • 1–19% | 440 (35) | 268 (33.9) | 172 (36.8) | |
| • 20–49% | 78 (6) | 51 (6.5) | 26 (5.6) | |
| • 50–79% | 77 (6) | 44 (5.6) | 29 (6.2) | |
| • ≥ 80% | 73 (6) | 46 (5.8) | 32 (6.8) | |
| Delayed graft function | 309 (24.6) | 116 (14.7) | 193 (41.2) | <0.0001 |
| T-cell depleting induction | 542 (43.1) | 265 (33.5) | 277 (59.2) | <0.0001 |
| Tacrolimus maintenance | 1043 (82.9) | 665 (84.2) | 378 (80.8) | 0.12 |

Table represents the same patient population as in a prior study reported by our group [20].

ESKD end-stage kidney disease; GN glomerulonephritis; PCKD polycystic kidney disease; CAKUT congenital anomalies of the kidneys and urinary tract; PRA panel reactive antibodies; CVD cardiovascular disease (either coronary artery disease, ischemic stroke, congestive heart failure, or atrial fibrillation).

## Infections

The most common infections were bacteremia, pneumonia and sepsis occurring in 18%, 15% and 10% of the study population respectively (Table 3). Throughout the study period, 423 (33.6%) participants had bacterial infection at a median (IQR) of 409 (53–1,654) days after transplant and at a median 124 (33–762) days after first RBCT among those who had bacterial

**Table 2. Transfusion characteristics.**

|  | Total cohort (N = 1,258) | Participants transfused (N = 468) |
|---|---|---|
| RBCT received; median (IQR) | 0 (0 to 2) | 3 (2 to 6) |
| Total amount of RBCT received; no. (%) |  |  |
| • None | 790 (62.8) | - |
| • 1 | 97 (7.7) | 97 (20.7) |
| • 2 | 118 (9.4) | 118 (25.2) |
| • 3–5 | 126 (10.0) | 126 (26.9) |
| • >5 | 127 (10.1) | 127 (27.1) |
| Time from transplant to 1st RBCT (days); median (IQR) | 0 (0 to 2.7) | 5.7 (2.1 to 72.2) |

Table represents the same transfusion data as in a prior study reported by our group [20].

RBCT red blood cell transfusion; IQR interquartile range.

infection occur after RBCT. The crude cumulative incidence of bacterial infection increased progressively during follow-up, reaching 59.2% for the whole study period (Fig 2). The study cohort for the analysis of the viral infection outcome (from March 14th, 2014 onwards) comprised of 452 individuals and 111 (24.6%) developed a viral infection at a median (IQR) of 154 (89–294) days after transplant and at a median 120 (55–318) days after first RBCT among those who had viral infection occur after RBCT. The cumulative incidence of viral infection was 30.7% for the entire follow-up period (Fig 2).

## Association of RBCT with infections

Among individuals transfused 1, 2, 3–5 and >5 RBC, compared to individuals never transfused, the adjusted HRs (95% CI) for bacterial infection were 1.35 (0.95 to 1.91), 1.29 (0.92 to

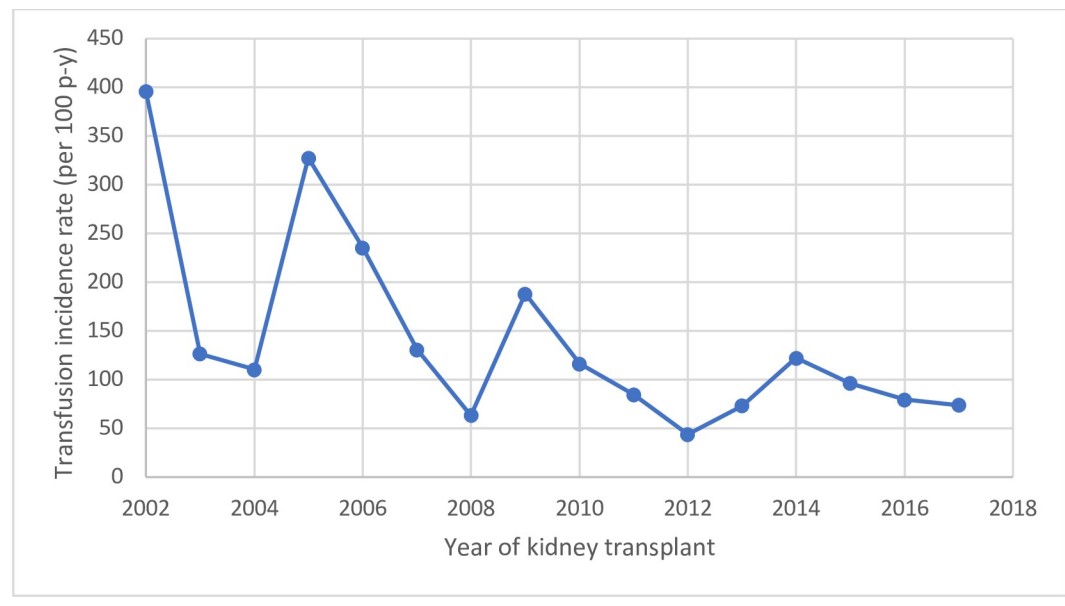

**Fig 1. One-year incidence rates (per 100 person-years) of RBCT by year of transplant.** For example, individuals receiving their kidney transplant in 2012 had a one-year incidence of blood transfusion of approximately 50 transfusions per 100 person-years of follow-up. Figure represents the same transfusion data as in a prior study reported by our group [20].

**Table 3. Types of infections occurring throughout the study period.**

|  | Total | Never transfused | Transfused during study | p-value |
|---|---|---|---|---|
| Bacteriemia | 222 (17.7) | 73 (9.2) | 149 (31.8) | <0.0001 |
| Pneumonia | 186 (14.8) | 79 (10.0) | 107 (22.9) | <0.0001 |
| Sepsis | 131 (10.4) | 46 (5.8) | 85 (18.2) | <0.0001 |
| Urinary tract infection | 77 (6.1) | 40 (5.1) | 37 (7.9) | 0.0421 |
| C Diff | 63 (5.0) | 22 (2.8) | 41 (8.8) | <0.0001 |
| BK nephropathy | 45 (3.6) | 22 (2.8) | 23 (4.9) | 0.0493 |
| BK viremia[a] | 79 (9.0) | 52 (8.0) | 27 (8.9) | 0.7240 |
| CMV tissue invasive disease | 19 (1.5) | 5 (0.6) | 14 (3.0) | 0.0009 |
| CMV viremia[b] | 52 (11.5) | 28 (9.2) | 24 (16.4) | 0.0232 |

[a] Data for BK viremia only available from 1 February 2007 onwards. % represents that of the corresponding study cohort (N = 960).

[b] Data for CMV viremia only available from 15 March 2014 onwards. % represents that of the corresponding study cohort (N = 452).

1.82), 2.63 (1.94 to 3.56) and 3.38 (2.30 to 4.95), respectively. For viral infections, compared to individuals never transfused, the adjusted HRs (95% CI) were 1.41 (0.80 to 2.47), 0.86 (0.40 to 1.82), 1.96 (1.03 to 3.74) and 1.06 (0.25 to 4.52), for RBCT 1, 2, 3–5 and >5 respectively (Table 4).

## Additional analyses

When accounting for various time-lags between RBCT and infection for the RBCT to count as an exposure, the HRs for bacterial infection attenuated, but remained significant for the highest levels of RBCT. For viral infection, the HRs remained non-significant (S4 Table). The association of RBCT with any infection (bacterial or viral) was similar to that seen for bacterial infection (S5 Table). When stratifying by whether or not DGF occurred, we found similar results (S6 Table). After randomly assigning 50% of study participants transfused on the day of surgery as having received their first RBCT immediately after transplant, re-analysis showed similar findings for all outcomes (S7 Table), as did re-analysis controlling for the year of transplantation (S8 Table). There were 197 acute rejection episodes among our study cohort and 51 infections occurred after rejection. Infection was more common among those who were transfused, whether infection occurred after a rejection or not (S9 Table). When restricting follow-up to the first 90-days after transplant, results were similar to those in our original analysis (S10 Table).

## Discussion

Among 1,258 kidney transplant recipients followed over a median of 3.8 years, we found a high infection risk (total cumulative incidence: bacterial infection 59.2%, viral infection 30.7%) and the receipt of RBCT was associated with bacterial, but not viral, infection in a dose-dependent manner (Bacterial infection: 1 RBC, HR 1.35; 2 RBC, HR 1.29; 3–5 RBC, HR 2.63; >5 RBC HR 3.38. Viral infection: 1 RBC, HR 1.56; 2 RBC, HR 0.86; 3–5 RBC, HR 1.96; >5 RBC, HR 1.06).

Our study confirms that kidney transplant patients are at high risk for developing various types of infections. Indeed, we found a cumulative incidence of 17% at 1-year and 31% at 5-years for bacterial infection. Furthermore, the majority of infections required hospital admission for diagnosis, suggesting heightened severity. This highlights the importance of identifying and limiting modifiable risk factors for infections.

A) Bacterial infections

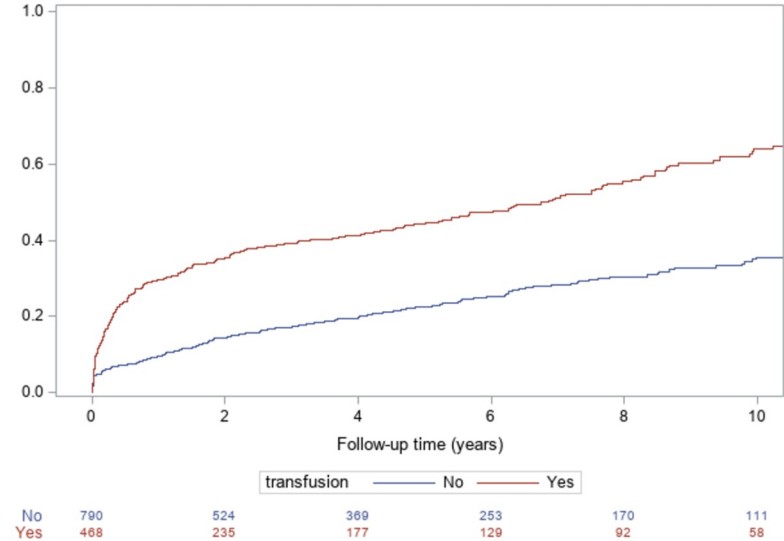

### Crude cumulative incidence of bacterial infection

|  | No. (%) | No events (%) | 1-year | 5-year | 10-year |
|---|---|---|---|---|---|
| Full cohort | 1,258 | 423 (33.6) | 16.9% | 30.7% | 47.0% |
| Transfusion No | 790 (62.8) | 189 (23.9) | 9.4% | 22.6% | 35.5% |
| Transfusion Yes | 468 (37.2) | 234 (50.0) | 29.5% | 44.3% | 63.9% |

B) Viral infections

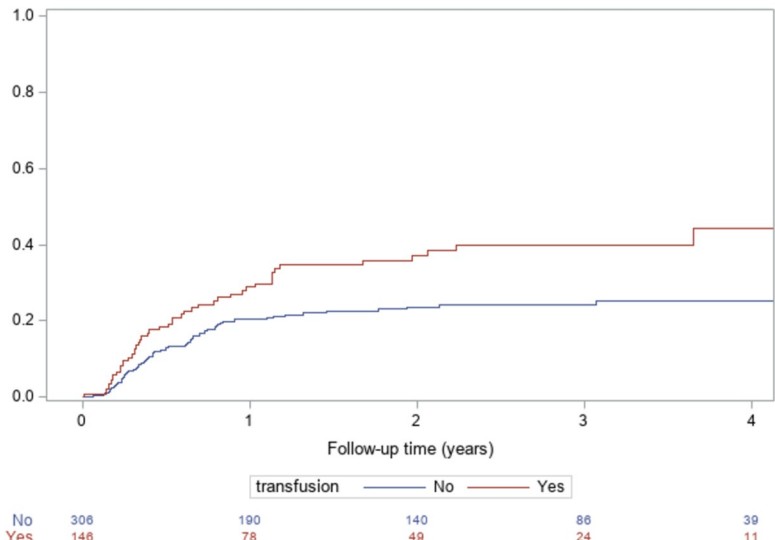

### Cumulative incidence of viral infections

|  | No. (%) | No. events (%) | 1-year | Full follow-up |
|---|---|---|---|---|
| Full cohort | 452 | 111 (24.6) | 23.0% | 30.7% |
| Transfusion No | 306 (67.7) | 64 (20.9) | 20.4% | 25.1% |
| Transfusion Yes | 146 (32.3) | 47 (32.2) | 28.9% | 44.2% |

**Fig 2. Kaplan-meier crude cumulative incidence curves for outcomes by transfusion status.** A) Bacterial infections.
B) Viral infections. Shown are the total number of infections, the 1, 5 and 10-year cumulative incidences as well as the
crude Kaplan-meier cumulative incidence curves by transfusion status.

**Table 4. Cox model HRs for infections based on cumulative RBCT exposure.**

| Outcome | # RBC units received | # events (%) | Time-varying crude HR (95% CI) | Time-varying adjusted HR (95% CI)* | p-value |
|---|---|---|---|---|---|
| Bacterial infection | None | 189 (23.9) | Reference | Reference | |
| | Any amount | 234 (50.0) | 2.14 (1.76 to 2.61) | 1.82 (1.48 to 2.24) | <0.0001 |
| | 1 | 37 (38.1) | 1.50 (1.06 to 2.12) | 1.35 (0.95 to 1.91) | 0.0949 |
| | 2 | 39 (33.1) | 1.40 (1.00 to 1.96) | 1.29 (0.92 to 1.82) | 0.1435 |
| | 3–5 | 67 (53.2) | 3.35 (2.51 to 4.45) | 2.63 (1.94 to 3.56) | <0.0001 |
| | >5 | 91 (71.7) | 4.03 (2.81 to 5.76) | 3.38 (2.30 to 4.95) | <0.0001 |
| Viral infection (BK or CMV) | None | 64 (20.9) | Reference | Reference | |
| | Any amount | 47 (32.2) | 1.63 (1.10 to 2.41) | 1.32 (0.87 to 2.02) | 0.1973 |
| | 1 | 17 (36.2) | 1.81 (1.07 to 3.08) | 1.41 (0.80 to 2.47) | 0.2309 |
| | 2 | 6 (14.6) | 1.00 (0.48 to 2.07) | 0.86 (0.40 to 1.82) | 0.6880 |
| | 3–5 | 15 (38.5) | 2.44 (1.32 to 4.49) | 1.96 (1.03 to 3.74) | 0.0405 |
| | >5 | 9 (47.4) | 1.26 (0.31 to 5.13) | 1.06 (0.25 to 4.52) | 0.9379 |

* Adjusted for age, sex, transplant type, cause of ESKD, PRA (as a continuous variable), presence of CVD, receipt of t-cell depleting induction and type of maintenance therapy.

We found that RBCT post-kidney transplant was associated with a significantly higher risk for infection. Within the non-transplant population, previous studies suggest receipt of a blood transfusion to be independently associated with infection. A meta-analysis from 2016 of 31 randomized controlled trials comparing a restrictive to liberal transfusion strategy did not reveal a decreased risk for infections (pneumonia, wound infection, sepsis) [28]. However, in 2014 a meta-analysis looking specifically at infectious risks associated with a restrictive versus liberal transfusion strategy found a significantly decreased risk for nosocomial infections with a restrictive transfusion strategy [29]. The different findings from these two meta-analyses may be due to a large 2015 trial in cardiac surgery showing no increased infectious risk with liberal compared to restrictive transfusion [30]. Unfortunately, no trials have included kidney transplant recipients. An observational study by Mazzeffi et al reported a nearly 9-fold higher odds for sepsis among transfused kidney transplant recipients compared to those non-transfused, while there was no association with surgical site infection [31]. The study was limited to intra-operative transfusion exposure and had only a small number of sepsis events (8). Our findings provide further insights into the association between RBCT and infections in kidney transplant patients. We examined a wide array of infections with a focus on clinically relevant infections. We identified bacterial infections almost exclusively during a hospital admission suggesting a high degree of severity. Therefore, our finding of a 2- to 3-fold higher infectious risk with RBCT transfusion post-transplant is highly salient given the frequency of transfusions administered and the population's susceptibility to infection. Most of this increased risk seemed to occur in the first year after transplant, which is when the vast majority of RBCT were given to our kidney transplant recipients and also when they are most immunosuppressed. This indicates a period of heightened risk clinicians should be mindful of. Since acutely infected kidney transplant recipients may develop anemia requiring a RBCT, we conducted additional time lag analyses to account for reverse causality. This showed findings consistent with our primary analysis. Also, transplant patients requiring RBCT may represent an inherently sicker population at greater risk of developing infection. Even after controlling for important confounders such as the use of T-cell depleting therapy and co-morbidities, we found similar findings. Furthermore, additional analyses stratified by the presence of DGF, which may herald a patient at higher risk for infection, and controlling for year of transplant, thus accounting for an era

effect, also showed similar findings. That being said, as for any observational study, we recognize the possibility for unmeasured confounding.

Our study presents novel findings in that we examined the occurrence of two types of viral infections of major clinical relevance to kidney transplant patients, BK and CMV infections, in relation to the receipt of RBCT. The mechanisms through which a RBCT may exert immunosuppressive effects are not fully understood. A RBCT may lead to suppression of lymphocyte function which increases as the number of transfusions increase, possibly due to immune suppressor cells contained in the transfusion [14, 16]. Furthermore, RBCT is implicated in inducing anergy of T-lymphocytes, a key immune modulator [17]. These alterations in lymphocytes and cellular immunity pathways by RBCT should, theoretically, lead to higher susceptibility to both viral and bacterial infections [32–34], a finding we did not observe in the current study. Possibilities for these seemingly discordant results include the smaller number of viral infections that underpowered our ability to detect a difference or the fact that BK and CMV infections are mostly diagnosed and managed in the ambulatory setting, making these less clinically severe than our bacterial infections. Conversely, bacterial infections were higher in a stepwise manner with RBCT. The association of RBCT with bacterial infections is consistent with non-transplant studies in multiple other post-operative settings including cardiovascular, colorectal cancer and orthopedics surgery [11, 15, 35–37], and the exact immune mechanism remains unclear.

The strengths of our study lie in its size and the capture of a diverse array of infections, including BK virus and CMV infections. We had an inclusive study design of all adult kidney transplant recipients at a large academic centre in Canada, which should make our results generalizable to other centres with similar kidney transplant populations. Also, we used a time-varying analysis which is important to limit survival bias when dealing with a baseline immeasurable exposure such as RBCT [38], we performed analyses to account for reverse causation and also taking into account as many confounders as possible.

Our study has limitations. First, we did not capture bacterial infections which are diagnosed and treated as an outpatient such as minor skin, respiratory tract or urinary tract infections. Therefore, our results are only applicable to serious bacterial infections likely to require a hospital admission. Second, being a retrospective study using routinely collected data, there are limitations inherent to the study design. We conducted a number of sensitivity analyses which showed consistent findings, including analyses accounting for reverse causation, for confounders and for possible misclassification of RBCT occurring on day 0. Third, since we only captured exposure and outcomes occurring at our centre, there may be missing data. However, our transplant population is instructed to present to one of the TOH campuses if they require urgent medical evaluation and if an individual were to require hospital admission for a severe infection, it is likely they would be transferred to TOH. Fourth, we did not ascertain the indication of RBCT, a potentially important confounder. Given that the vast majority of RBCT occurred in the first week post-transplant, we suspect most were given simply for low hemoglobin values in the context of delayed graft function. Our analyses stratified by DGF status showed similar findings. Finally, we did not have information on doses or levels of immunosuppressants at time of infection. This could be problematic since over-immunosuppression increases the risk for infection and anemia. That being said, our patients are nearly all treated with the same immunosuppressants (tacrolimus, mycophenolate and prednisone) with mycophenolate 1g twice a day, tacrolimus trough of 4–6 ng/mL from 6 months onwards and prednisone 5mg daily from 3 months onwards. Doses may be reduced after a severe infection, however in the present study this would occur after the outcome and therefore we believe the majority of patients would have been on the same doses of medications at time of infection.

In conclusion, we found that receipt of RBCT post-kidney transplant was consistently and in a stepwise manner associated with higher bacterial infection with no elevation in viral (BK

or CMV) infection risk. Given the high rates of infection in kidney transplant patients and the frequency with which they are exposed to transfusions, receipt of RBCT may represent an important, modifiable risk factor for infection and attempts should be made to transfuse the minimal number of units clinically required.

## Supporting information

**S1 File. Usual immunosuppression protocol for kidney transplant at TOH.**
(DOCX)

**S1 Table. RECORD guidelines checklist.**
(DOCX)

**S2 Table. ICD10 codes used for identifying infections.**
(DOCX)

**S3 Table. Baseline characteristics ascertained for the study.**
(DOCX)

**S4 Table. Association of RBCT with outcomes for different time-lags between exposure and occurrence of outcome (HR [95% CI]).**
(DOCX)

**S5 Table. Risks for any infection (bacterial or viral) by cumulative RBCT exposure.**
(DOCX)

**S6 Table. Time-varying, adjusted hazard ratios (95% CI) for infections, stratified by DGF status of kidney transplant.**
(DOCX)

**S7 Table. Association of RBCT with outcomes accounting for RBCT occurring on day of transplant surgery.**
(DOCX)

**S8 Table. Time-varying, adjusted hazard ratios (95% CI) for outcomes, controlling for year of transplantation.**
(DOCX)

**S9 Table. Transfusion frequency among those with infection, whether the infection occurred after a rejection or not.**
(DOCX)

**S10 Table. Time-varying, adjusted hazard ratios (95% CI) for outcomes, when restricting follow-up to the first 90-days post-transplant.**
(DOCX)

**S1 Fig. Mean hemoglobin level pre-transfusion by year of transfusion.**
(DOCX)

## Author Contributions

**Conceptualization:** David Massicotte-Azarniouch, Manish M. Sood, Dean A. Fergusson, Alan Tinmouth, Greg A. Knoll.

**Data curation:** David Massicotte-Azarniouch.

**Formal analysis:** David Massicotte-Azarniouch.

**Methodology:** David Massicotte-Azarniouch, Manish M. Sood, Dean A. Fergusson, Michaël Chassé, Greg A. Knoll.

**Supervision:** Manish M. Sood, Greg A. Knoll.

**Writing – original draft:** David Massicotte-Azarniouch.

**Writing – review & editing:** David Massicotte-Azarniouch, Manish M. Sood, Dean A. Fergusson, Michaël Chassé, Alan Tinmouth, Greg A. Knoll.

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
