## [Decision Letter · Decision Letter 0]

19 Aug 2021

PONE-D-21-19259

Blood transfusion and the risk for infections in kidney transplant patients

PLOS ONE

Dear Dr. Massicotte-Azarniouch,

Thank you for submitting your manuscript to PLOS ONE. After careful consideration, we feel that it has merit but does not fully meet PLOS ONE’s publication criteria as it currently stands. Therefore, we invite you to submit a revised version of the manuscript that addresses the points raised during the review process.

ACADEMIC EDITOR:

This is an interesting paper on a clinically relevant topic. Two expert reviewers have commented on the MS and have reommended revisions. This is largely due to th need for further clarifications on the topics specified below. I think the crucial question that needs to be answered (supported by data) is whether the association between RBCT and bacterial infections is a surrogate for KTx patients that have significant co-morbidities and/or complications and thus will anyway be more susceptible to infections?

We look forward to receiving your revised manuscript.

Kind regards,

Frank JMF Dor, M.D., Ph.D., FEBS, FRCS

Academic Editor

PLOS ONE

Journal Requirements:

3. Thank you for stating in the text of your manuscript "Due to the retrospective nature of our study and the use of deidentified data, informed consent was waived.". Please also add this information to your ethics statement in the online submission form.

4. Please include your actual numerical p-values in Table 4

7.  Thank you for submitting the above manuscript to PLOS ONE. During our internal evaluation of the manuscript, we found significant text overlap between your submission and the following previously published work, of which you are an author. 

- https://www.kireports.org/article/S2468-0249(21)00017-6/fulltext

We would like to make you aware that copying extracts from previous publications, especially outside the methods section, word-for-word is unacceptable. In addition, the reproduction of text and data from published reports has implications for the copyright that may apply to the publications.

Please revise the manuscript to rephrase the duplicated text, cite your sources, and provide details as to how the current manuscript advances on previous work. Please note that further consideration is dependent on the submission of a manuscript that addresses these concerns about the overlap in text with published work.

We will carefully review your manuscript upon resubmission, so please ensure that your revision is thorough

Reviewers' comments:

Reviewer's Responses to Questions

**Comments to the Author**

1. Is the manuscript technically sound, and do the data support the conclusions?

Reviewer #1: Yes

Reviewer #2: Yes

2. Has the statistical analysis been performed appropriately and rigorously? 

Reviewer #1: Yes

Reviewer #2: Yes

3. Have the authors made all data underlying the findings in their manuscript fully available?

Reviewer #1: Yes

Reviewer #2: Yes

4. Is the manuscript presented in an intelligible fashion and written in standard English?

Reviewer #1: Yes

Reviewer #2: Yes

5. Review Comments to the Author

Reviewer #1: I wish to compliment the authors with regard to their manuscript.

As a whole I consider this to be a comprehensive paper.

However I am confused with regard to the presented analysis. If I understand the manuscript correctly the authors focused extensively upon the question if early infections in relation to transfusion could be two signs of the same underlying cause and therefor there might reflect some bias.

However I can't find a proper evaluation between the timing of the transfusion and the timing of the first endpoint. If I understand the analysis correctly then even ten years after transfusion there appears to be a higher incidence of bacterial infection.

I would like more information on the time relation between transfusion and the occurrence of the first end point. What is the median/ mean duration in weeks or days. One would expect for an immunomodulatory effect of a transfusion to occur and affect the chances of infection; too short an interval does not feel likely however too long an interval seems equally unlikely; one would assume that such an effect would disappear over time.

If this risk indeed persists over ten years as I comprehend from the manuscript I have two questions for the authors:

- If a higher rate of infection can be evaluated so long after transfusion then in the analysis some attention should be paid to transfusions prior to transplantation; as they might affect the observed incidences (assuming that those transfusions also affect the chance of infections for years).

- If a higher rate of infection can be evaluated so long after transfusion I would like to invite the authors to propose an explanation for this phenomenon

In examining figure 2 one could also propose that two years after transfusion the (annual) rate of infection between transfused and non-transfused patients actually appears to be similar and that the effect of transfusion on infection rate actually is a temporary effect which seems to disappear over time.

Taking this into account one could wonder if a ten year follow-up for the analyses is appropriate. Some reflection on the timing of achievement of the endpoint after transfusion is required.

Reviewer #2: Thank you for asking me to review this submission from Massicotte-Azarniouch and colleagues exploring use of red blood cell transfusion ater kidney transplantation and risk for bacterial/viral infection, finding a positive association with bacterial infections only. The mansucript is interesting, well written and deals with an important issues.

My comments t help better understand the submission and aid revisions are:

1. On page 8 the authors state 'DGF is considered high-immunological risk at our

196 institution and such patients receive thymoglobulin'. How are people at risk for DGF identified prior to transplant? ECD kidneys? DCD kidneys?

2. The statistical analyses are comprehenisve. The authors mention they censored for death/graft loss but I wonder whether they should consider doing a competing risk analysis? Clearly a kidney transplant patient who dies or loses their kidney is no longer at risk of exposure to the outcome of interest.

3. Does TOH have a pre-operative policy for threshold Hb levels for transplant surgery to proceed?

4. In Table 1 the authors display the tacrolimus use (which I assume is time depednent with their eriod of study with most use after 2007) but what is the protocol for immunosuppression? Which anti-proliferatives are used and how are corticosteroids used at the center? I see this nformation is mentioned towards the end of the discussion but I think burden of immunosuppression is important to know.

5. How would any blood transfusion and/or infection outside of TOH be captured?

6. What CMV prophylaxis is used at TOH? Is CMV and/or BK routinely checked post-transplant at certain time points?

7. Do the auhtors think the association between incremental RBCT and bacterial infections is a surrogate for a kidney transplant patient with significant co-morbidity and/or complications that will render them susceptible to infections anyway? It may be hard to extract this data but any chance of knowing what the indication for giving RBCT was as this is likely to be a major confounder (and should be cited as such if this cannot be analysed).

8. The authors cite 2 meta-analyses on page 15 of the discussion with diametrically different outcomes - how were these two studies different to arrive at such different conclusions?

6. PLOS authors have the option to publish the peer review history of their article (what does this mean?). If published, this will include your full peer review and any attached files.

Reviewer #1: No

Reviewer #2: No

---

## [Author Response · Author response to Decision Letter 0]

17 Sep 2021

Please see attached/uploaded document with the response to the reviewers' questions and comments

---

## [Decision Letter · Decision Letter 1]

18 Oct 2021

Blood transfusion and the risk for infections in kidney transplant patients

PONE-D-21-19259R1

Dear Dr. Massicotte-Azarniouch,

We’re pleased to inform you that your manuscript has been judged scientifically suitable for publication and will be formally accepted for publication once it meets all outstanding technical requirements.

Kind regards,

Frank JMF Dor, M.D., Ph.D., FEBS, FRCS

Academic Editor

PLOS ONE

Additional Editor Comments (optional):

Reviewers' comments:

Reviewer's Responses to Questions

**Comments to the Author**

1. If the authors have adequately addressed your comments raised in a previous round of review and you feel that this manuscript is now acceptable for publication, you may indicate that here to bypass the “Comments to the Author” section, enter your conflict of interest statement in the “Confidential to Editor” section, and submit your "Accept" recommendation.

Reviewer #2: All comments have been addressed

2. Is the manuscript technically sound, and do the data support the conclusions?

Reviewer #2: Yes

3. Has the statistical analysis been performed appropriately and rigorously? 

Reviewer #2: Yes

4. Have the authors made all data underlying the findings in their manuscript fully available?

Reviewer #2: No

5. Is the manuscript presented in an intelligible fashion and written in standard English?

Reviewer #2: Yes

6. Review Comments to the Author

Reviewer #2: Thank you for your revisions and responses to my queries. I have no further comments from my perspective.

7. PLOS authors have the option to publish the peer review history of their article (what does this mean?). If published, this will include your full peer review and any attached files.

Reviewer #2: No

---

## [Editor Report · Acceptance letter]

2 Nov 2021

PONE-D-21-19259R1 

Blood transfusion and the risk for infections in kidney transplant patients 

Dear Dr. Massicotte-Azarniouch:

I'm pleased to inform you that your manuscript has been deemed suitable for publication in PLOS ONE. Congratulations! Your manuscript is now with our production department. 

Kind regards, 

on behalf of

Dr. Frank JMF Dor 

Academic Editor

PLOS ONE